# Combined analyses of RNA-sequence and Hi-C along with GWAS loci—*A novel approach to dissect keloid disorder genetic mechanism*

Jia Huang[1,2¤], Xiaobo Zhou[1,2], Wenbo Wang[1,2], Guangdong Zhou[1,2,3], WenJie Zhang[1,2,3], Zhen Gao[1], Xiaoli Wu[1], Wei Liu[1,2,3]*

**1** Department of Plastic and Reconstructive Surgery, Shanghai Ninth People's Hospital, Shanghai Jiao Tong University School of Medicine, Shanghai, China, **2** Shanghai Key Laboratory of Tissue Engineering Research, Shanghai, China, **3** National Tissue Engineering Centre of China, Shanghai, China

¤ Current address: Department of Dermatology, Huashan Hospital, Fudan University, Shanghai, China
* liuwei_2000@yahoo.com

## Abstract

Keloid disorder is a tumour-like disease with invasive growth and a high recurrence rate. Genetic contribution is well expected due to the presence of autosomal dominant inheritance and various genetic mutations in keloid lesions. However, GWAS failed to reveal functional variants in exon regions but single nucleotide polymorphisms in the non-coding regions, suggesting the necessity of innovative genetic investigation. This study employed combined GWAS, RNA-sequence and Hi-C analyses to dissect keloid disorder genetic mechanisms using paired keloid tissues and normal skins. Differentially expressed genes, miRNAs and lncRNAs mined by RNA-sequence were identified to construct a network. From which, 8 significant pathways involved in keloid disorder pathogenesis were enriched and 6 of them were verified. Furthermore, topologically associated domains at susceptible loci were located via the Hi-C database and ten differentially expressed RNAs were identified. Among them, the functions of six molecules for cell proliferation, cell cycle and apoptosis were particularly examined and confirmed by overexpressing and knocking-down assays. This study firstly revealed unknown key biomarkers and pathways in keloid lesions using RNA-sequence and previously reported mutation loci, indicating a feasible approach to reveal the genetic contribution to keloid disorder and possibly to other diseases that are failed by GWAS analysis alone.

## Author summary

Keloid disorder is a benign skin tumour characterized by uncontrolled fibroproliferative tissue growth, which only occurs in human beings with severe reoccurrence post-therapy. It affects several hundred million people with difficulty to control its growth and relapse. It has been long thought that exonic gene mutations must play an important role, but large-scaled GWAS analyses only revealed 3 single nucleotide polymorphisms in the non-coding regions as previously reported. For the first time, this study demonstrated that the

**Data Availability Statement:** Datasets related to this article can be found at https://dataview.ncbi. nlm.nih.gov/object/PRJNA736745?reviewer= okvm6kr2108v7s7su502kqfkts, hosted at the U.S.

National Institutes of Health's National Library of Medicine (NIH/NLM). All other data generated or analyzed during this study are included in this published article and in the supplementary files.

**Funding:** This research was supported by the National Natural Science Foundation (81671921 to WL). The funders had no role in study design, data collection and analysis, decision to publish, or preparation of the manuscript.

**Competing interests:** The authors have declared that no competing interests exist.

true genetic mechanism is likely to be the dysfunctional epigenetic regulation caused by mutations in regulatory elements at the non-coding region as revealed by the combined analyses of GWAS, RNA-sequence and Hi-C data. This approach may lead to the breakthrough of keloid disorder genetic/epigenetic mechanism, if further large-scaled analyses are performed along with human keloid tissue Hi-C data.

## Introduction

Keloid disorder is considered as a benign skin tumour that exhibits many cancer-like characteristics such as uncontrollable and invasive growth with an extremely high recurrence rate. Unlike hypertrophic scar, keloid lesions are lack of spontaneous regression and able to cross the original wound boundary. Keloid lesions usually appear as highly vascularized, rubbery and fibrous nodules in their gross view and typically occur in response to surgery, lacerations, burns or inflammatory skin conditions. The pathogenesis of keloid disorder is complicated and involves uncontrolled dermal fibroblast proliferation, excessive deposition of extracellular matrices (ECM) including collagen and proteoglycans, increased inflammatory cell infiltration and viciously dysregulated cytokines [1]. However, gene mutations are increasingly recognized as one of the critical risk factors for keloid lesion formation.

Previously, genetic investigation usually focused on exon mutation using genome-wide association study (GWAS) in order to explore the potential pathogenic gene mutation of functional molecules. However, GWAS literature only revealed some associated risk single nucleotide polymorphisms (SNPs) in the non-coding regions instead of identifying specific susceptibility genes and pathogenic signalling pathways in keloid lesions [2,3]. These phenomena suggest that a new strategy might be needed to further decipher the genetic contribution to keloid disorder pathogenesis. GWAS usually requires a large sample size to qualify the assay. Instead, the widely applied RNA-sequence (RNA-seq) technique can efficiently generate reliable data with a relatively small sample size, particularly for self-controlled tissue samples [4–7].

Genetic and epigenetic abnormalities are now believed to be closely implicated in tumour initiation and progression. It has been proven that mutation in the non-coding regions could result in dysregulated expression of microRNAs (miRNAs) and long non-coding RNAs (lncRNAs), which in turn affects the expression of protein-coding genes (mRNAs). For example, overexpression of lncRNA NR2F1-AS1 was shown to promote the proliferation and migration of thyroid cancer cells through regulating the miR-338-3p/CCND1 axis [8]. Downregulated miR-338-3p could accelerate hepatocellular carcinoma cell growth through targeting FOXP4 [9]. All those aberrations in mRNA, miRNA and lncRNA could directly or indirectly regulate protein expression at post-transcription level.

Because of this reason, this study investigated the genetic and epigenetic differences between keloids keloid tissues and their surrounding normal skins using RNA-seq analyses of mRNA, miRNA and lncRNA combined with the analysis of published GWAS data in order to mine dysregulated gene expression and related signalling pathways. In addition, high-throughput chromosome conformation capture (Hi-C) was utilized to analyse the susceptible chromosome loci of keloid lesions that were reported previously [2,3], and Hi-C was also combined with RNA-seq for the correlation analysis and functional verification (See Fig 1), which may set as an example of a new strategy to dissect the genetic mechanism of keloid lesion formation and development.

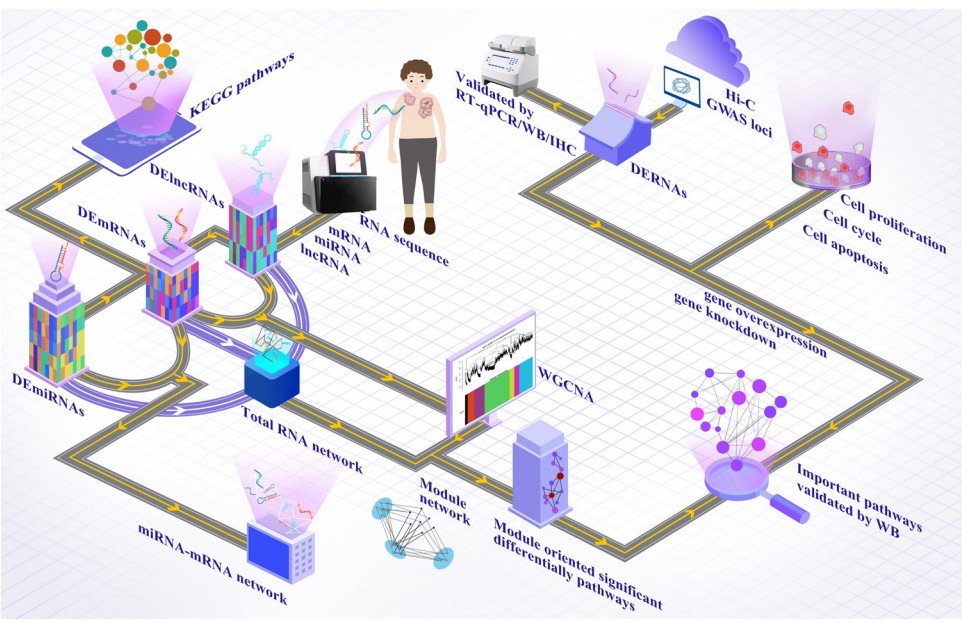

**Fig 1. The flow chart of the experiment.**

## Results

### Differential RNA expression between keloid tissues and their adjacent normal skins

As shown in Fig 2A, the keloid lesions appear as unsightly firm, rubbery, bosselated and outgrowing tumours with pink and purple colours. The differential expressed mRNAs (DEGs), lncRNAs (DElncRNAs) and miRNAs (DEmiRNAs) between keloid lesions and their corresponding adjacent normal skins (uniformly detected in all 3 paired samples, K1-K3, S1 Table) were calculated from the RNA-seq data to form a heatmap (S1 Fig). There were total 3735 DEGs consisting of 1022 upregulated and 2713 downregulated; 3074 DElncRNAs consisting of 1172 upregulated and 1902 downregulated; 270 DEmiRNAs consisting of 143 upregulated and 127 downregulated (S3–S5 Tables). Then the Kyoto Encyclopedia of Genes and Genomes (KEGG) enrichment was analysed on all three types of DERNAs.

As shown in Fig 2B, 14 significantly enriched KEGG pathways were identified including Wnt, Tyrosine metabolism, Retinol metabolism, peroxisome proliferator-activated receptor (PPAR), Pancreatic secretion, Neuroactive ligand-receptor interaction, Metabolism of xenobiotics by cytochrome P450, Inflammatory mediator regulation of TRP channels, Hippo, Fatty acid degradation, Estrogen, ECM-receptor interaction, Cell adhesion molecules (CAMs) and Signalling pathways regulating pluripotency of stem cells. All the DEGs and DElncRNAs were then analysed using weighted gene co-expression network analysis (WGCNA) and the weighted co-expression network was modelled to divide RNAs into 7 modules (S2 Fig) as previously described [10]. Among them, three modules (blue, red and turquoise) were significantly differentially expressed between keloid lesions and adjacent normal skin samples (Fig 2C).

### The RNA regulation network

The entire RNA regulation network was built on the differentially expressed RNAs. A total of 52556 DEmiRNA-target (mRNA) interaction pairs consisting of 270 DEmiRNAs and 12221

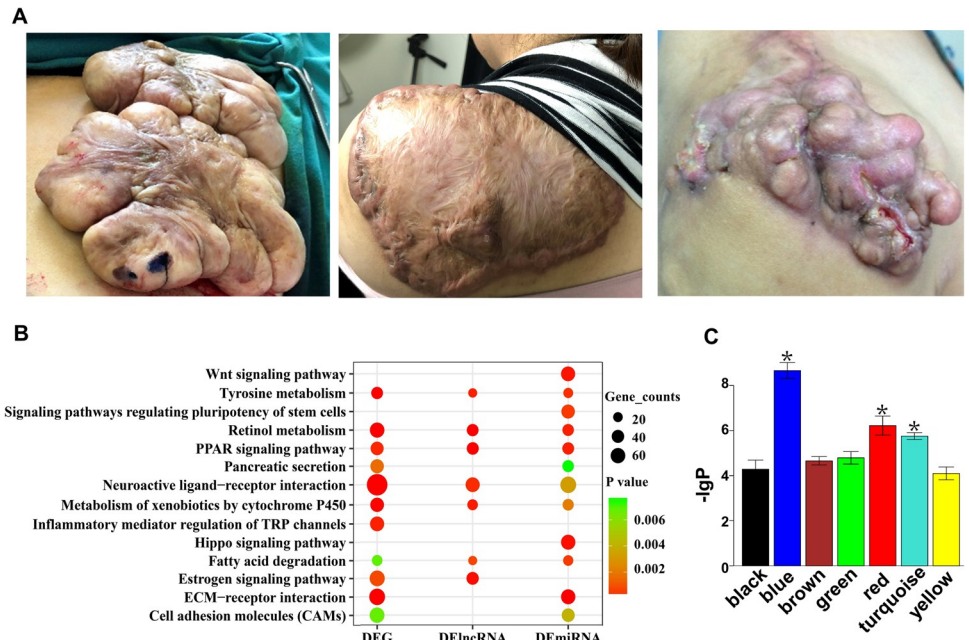

**Fig 2. Keloid lesion gross appearance and enriched KEGG pathways and WGCNA (weighted gene co-expression network analysis) identified modules.** (A) Gross appearance of three selected keloid lesion samples. (B) Identification of the differentially expressed signalling pathways between keloid lesions and adjacent normal skins by KEGG enrichment of DERNAs (mRNA, miRNA and lncRNA-targeted mRNA). (C) Seven gene co-expression modules represented by 7 different colors were detected by WGCNA analysis of the paired DEG and DElncRNA and the modules of blue, red and turquoise showed most significantly differentially expressed between keloid lesions and adjacent normal skin samples.

targeting genes were constructed (S3 Fig) and the whole miRNA targeted genes were shown in S6 Table. The topological characteristics of the top 20 nodes in the network were displayed in S7 Table. Among them, hsa-miR-335-5p, which was downregulated in keloid tissues, showed most targets including ITGB8 (Integrin beta-8, a receptor for fibronectin which could promote keloid lesion formation) [11], PARD6B (par-6 family cell polarity regulator beta, which was involved in cell division and cell polarization) [12], RUNX2 (Runt-related transcription factor 2, which could transactivate collagen I and collagen X) [13], WDR17 (WD Repeat Domain 17, associated with retinal disease) [14], LONRF2 (LON Peptidase N-Terminal Domain And Ring Finger 2, which was upregulated in pancreatic ductal adenocarcinoma) [15], XKR4 (XK Related 4, which could affect cerebellar development) [16], RRAGD (Ras-related GTP binding D, which could directly activate the mTOR signalling pathway and subsequently inhibit the autophagy) [17], FMNL3 (Formin-like protein 3, which was linked to oncogenic pathway) [18], LIFR (Leukaemia inhibitory factor receptor, which promoted tumour angiogenesis by up-regulating IL-8 levels) [19], RAB11FIP1 (RAB11 Family Interacting Protein 1, which had a ras-activating function and promoted cancer) (Fig 3A) [20].

Then the DEGs and DElncRNAs were mapped to the DEmiRNA-target networks while the protein-protein interactions were also imported to construct the total RNA regulation network (S4 Fig). And the three modules mentioned above were selected in the total network to further construct a fine regulatory network (S5 Fig). It was shown that the interactions between blue and turquoise modules were of tighter association (S5 Fig). As seen in Fig 3B, whole DEG-DE-miRNA and DEG-DElncRNA interaction pairs from the three WGCNA modules were used to construct the RNA interaction network. The interaction network contained several small sub-networks instead of a single big one (Fig 3B). Enrichment analysis was adopted to illuminate

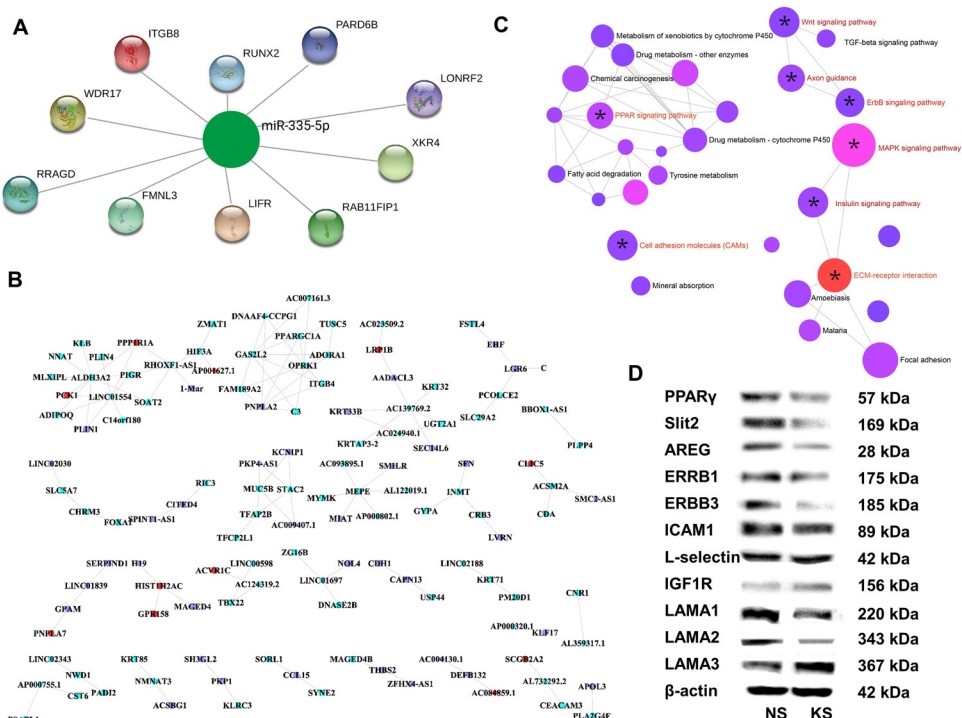

**Fig 3. Construction of DERNAs network and verification of enriched signalling pathways.** (A) Downregulated hsa-miR-335-5p and its top 10 targeting genes derived from DEmiRNA-target interaction network. (B) RNA interaction network constructed by DEG-DEmiRNA and DEG-DElncRNA interaction pairs from the three WGCNA modules. (C) Identified eight signal pathways enriched from the three modules of the RNA network. (D) Western blot verification of enriched differential signalling pathways between paired keloid tissues and adjacent normal skins.

the potential functions of the three modules and 8 signalling pathways including ECM-receptor interaction, CAMs, epidermal growth factor receptor (EGFR/ERBB), axon guidance, mitogen-activated protein kinase (MAPK), Wnt, insulin and PPAR pathway. They are present in the networks according to the top 8 P value (Fig 3C, labelled with*). To further explore the potential DERNAs involved in the 8 signalling pathways, these pathways were also mapped to the total RNA regulation network (S4 Fig) and a few nodes were represented (S6 Fig), which can be used for further needed analysis. These results suggested that epigenetic modification in keloid lesions can have a significant influence on keloid disorder pathogenesis via affecting the functions of important signalling pathways.

## The validation of dysregulated critical signalling pathways in keloid lesions

The pathogenic contributions of MAPK and Wnt pathways to keloid lesions have long been recognized [21,22], which indicated that the accuracy and quality of the sequencing data were sufficient for further analysis. The remaining 6 pathways mentioned above were then verified in the keloid and normal skin tissues by western bolt (WB) assay (Fig 3D). PPAR and axon guidance pathways might be inhibited in keloid lesions because of the downregulated expression of PPARγ and Slit2, respectively. ERBB pathway was found repressed due to the reduced expression of amphiregulin (AREG), ERBB1 and ERBB3. Slightly differentially expressed intercellular cell adhesion molecule 1 (ICAM1) and L-selectin implied the similar activity of CAMs pathway in keloid lesions and normal skin tissues. The insulin signalling pathway was activated in keloid lesions with highly upregulated insulin-like growth factor 1 receptor (IGF1R). Regarding the ECM-receptor interaction pathway, the downregulated laminin

subunit alpha 1 (LAMA1) and LAMA2 but upregulated LAMA3 were found in keloid lesions compared with those of normal skin tissues.

## DERNAs in the topologically associating domains (TADs) of susceptible loci

Gene expression could be modulated via changing the local chromatin structure, which directly links to chromatin organization and transcriptional network architecture. The genes in the same TADs share the same cis-regulatory regions with promoters of specific genes [23]. As shown in Fig 4A–4C, the TADs (red box) in the susceptible loci were retrieved from the IMR90 Hi-C database to detect the DERNAs. After mapping Hi-C data within the susceptibility locus to keloid tissue RNA-seq data, the RNA expressions within a particular locus were shown in Fig 4D–4F and S8 Table. The significantly differentiated RNAs were then selected based on the criteria of P value < 0.05 and |log2FC| ≥ 1 (Fig 4G). In locus 1q41, *HHIPL2 (hedgehog interacting protein-like 2), LINC01705, LINC01655* and *LINC02257* were significantly upregulated (Fig 4D and 4G). However, no RNAs were found to be significantly differentially expressed within locus 3q22.3 (Fig 4E and 4G). In locus 15p21.3, there were 16 RNAs exhibiting a constant downward trend as shown in S8 Table and Fig 4F and *TEX9 (testis-expressed 9), AC011912.1, DNAAF4-CCPG1* were significantly downregulated (Fig 4F and 4G). Unexpectedly, a few upregulated genes were observed within the sub-TAD of this locus (yellow box) and some significantly upregulated genes were found including *PYGO1 (pygopus family PHD finger 1), NEDD4 (neuronal precursor cell expressed developmentally downregulated 4)* and *RAB27A (ras-related protein rab-27a)* (Fig 4C, 4F and 4G).

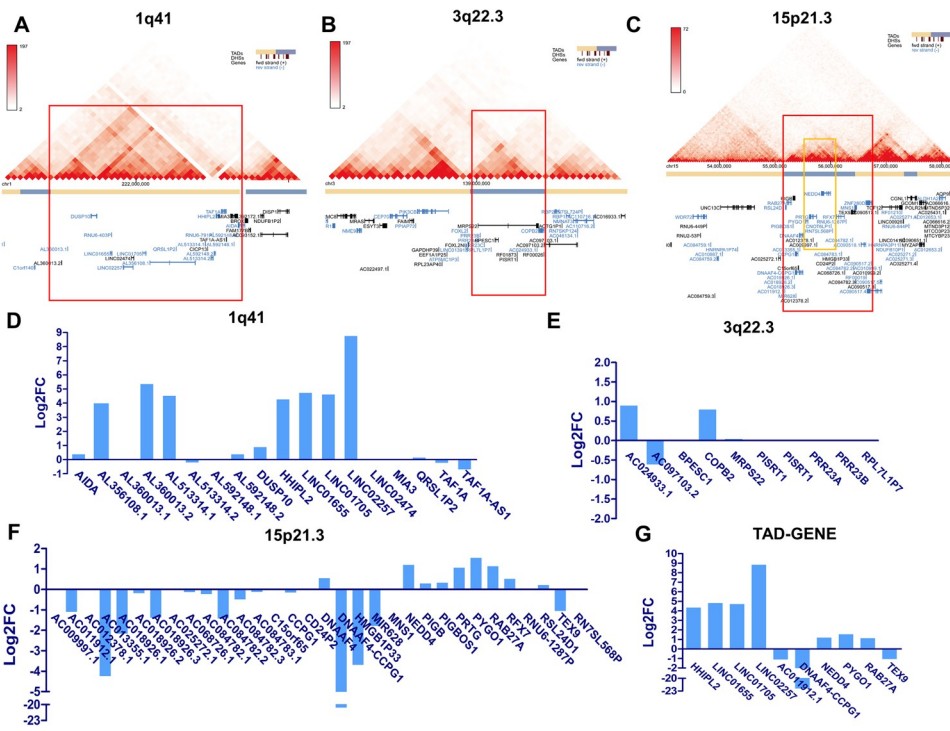

**Fig 4. Differentially expressed RNAs in TADs of susceptibility loci.** Differentially expressed RNAs in TAD (red outline) of locus 1q41 (A), locus 3q22.3 (B) and locus 15p21.3 (C). Detailed differentially expressed RNAs were listed in TAD of locus 1q41 (D), locus 3q22.3 (E) and locus 15p21.3 (F). Total ten significantly differentially expressed mRNAs/lncRNAs were obtained by screening through DERNAs with P < 0.05 and |log2FC| ≥ 1 (G).

These results are in agreement with the fact that loci 1q41 and 15p21.3 were susceptible for keloid disorder in Chinese Han population. Although there are SNP mutations at locus 3q22.3 in Japanese, but not in Chinese Han people [3], no significant DERNAs were found in this locus using tissue samples of Chinese keloid patients (Fig 4). Thus, the significant DEGs and DElncRNAs within the two TADs (1q41 and 15p21.3, Fig 4G) were selected for further validation of their expression and functions.

## The validation of selected biomarkers in keloid tissues

Immunohistochemistry (IHC), Western blot (WB) and real-time quantitative polymerase chain reaction (RT-qPCR) analysis showed that the selected four TAD genes (*HHIPL2*, *NEDD4*, *PYGO1*, *RAB27A*) and corresponding coding proteins were highly upregulated in keloid lesions, and *TEX9* and corresponding coding proteins validated by IHC were obviously downregulated in keloid lesions (Fig 5). Unexpectedly, TEX9 was upregulated in WB. As for the no-consistent result of WB and IHC, it is possibly because the same protein may not have the same conformation in SDS-PAGE used in WB and in the paraffin section usually used in IHC [24]. Additionally, the expressions of three selected lncRNAs (*LINC02257*, *LINC01705*, *LINC01655*) were significantly upregulated and two lncRNAs (*AC011912.1* and D*NAAF4-CCPG1*) were obviously downregulated in the keloid lesion samples than those in the normal skin samples (Fig 5D, P < 0.05). All the findings supported the high prediction accuracy of the sequencing and bioinformatics analysis.

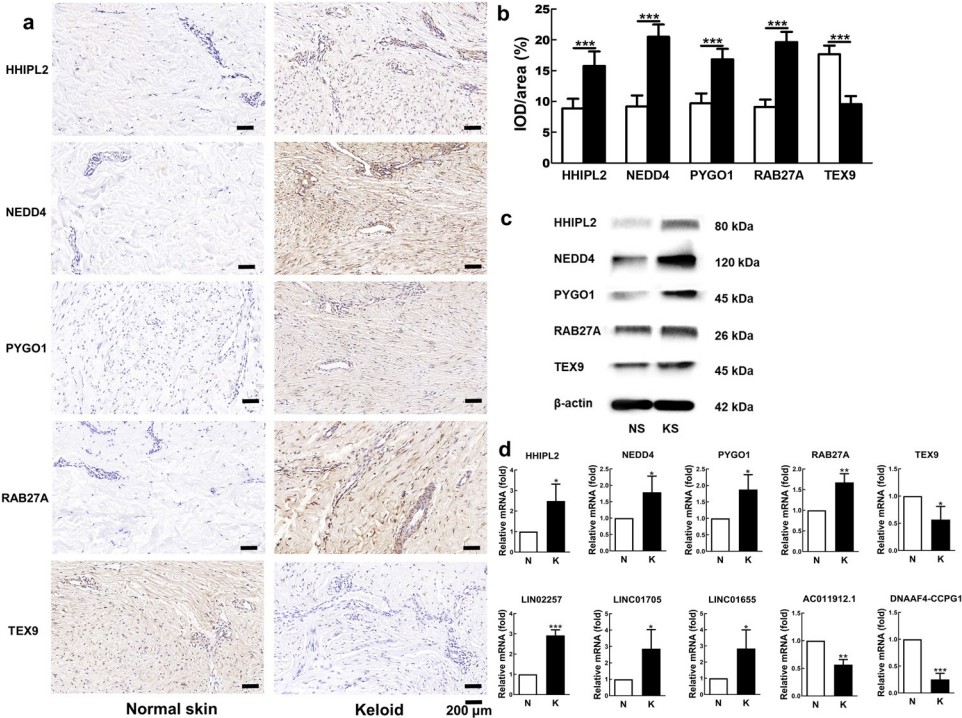

**Fig 5. Verification of DEGs/DElncRNAs *in vitro*.** Immunohistochemical staining (A) as well as its semi-quantitative analysis (B) and western blot (C) were used to detect the expression levels of HHIPL2, NEDD4, PYGO1, RAB27A and TEX9 in keloid tissues and their adjacent normal skins. Scale bars = 200 μm, magnification×200. RT-qPCR (D) was used to detect the gene expression levels of *HHIPL2*, *NEDD4*, *PYGO1*, *RAB27A*, *TEX9* and *LINC02257*, *LINC01705*, *LINC01655*, *lncRNAAC011912.1*, *DNAAF4-CCPG1* of keloid tissues and their adjacent normal skins. Each assay was repeated in at least three independent samples (respectively from KT1-KT5 samples, * P < 0.05, ** P < 0.01, *** P < 0.001).

## The effect of gene transfection on keloid fibroblasts (KFs) proliferation, cell cycle distribution and apoptosis

PYGO1, RAB27A and LINC02257 shRNA transfection could significantly decrease KFs viability compared with that of control cells at the time points of day 3 and day 5 post-transfection. Similarly, overexpression of TEX9, hsa-miR-335-5p and NDAAF4-CCPG1 could significantly diminish cell viability at day 3 and day 5 post-transfection (Fig 6A, P < 0.05).

Furthermore, knockdown of PYGO1, RAB27A and LINC02257 induced an obvious GO/G1 phase cell cycle arrest in KFs compared with the control cells as shown in Fig 6B (P < 0.05). A similar phenomenon was observed in the KFs with overexpression of TEX9, hsa-miR-335-5p and DNAAF4-CCPG1 (Fig 6B, P < 0.05).

Moreover, KFs transfected with PYGO1, RAB27A and LINC02257 shRNA and overexpressed of TEX9, hsa-miR-335-5p and NDAAF4-CCPG1 displayed a significant increase in early, late and total apoptosis in KFs compared to the cells of the control group (Fig 6C, P < 0.05). Representative histograms of cell cycle phase and cell apoptosis distribution were shown in S7 and S8 Figs.

## Discussion

Keloid disorder is a benign skin tumour characterized by fibroproliferative skin disorder that only occurs and relapses in humans with difficulty to control its growth and invasion into

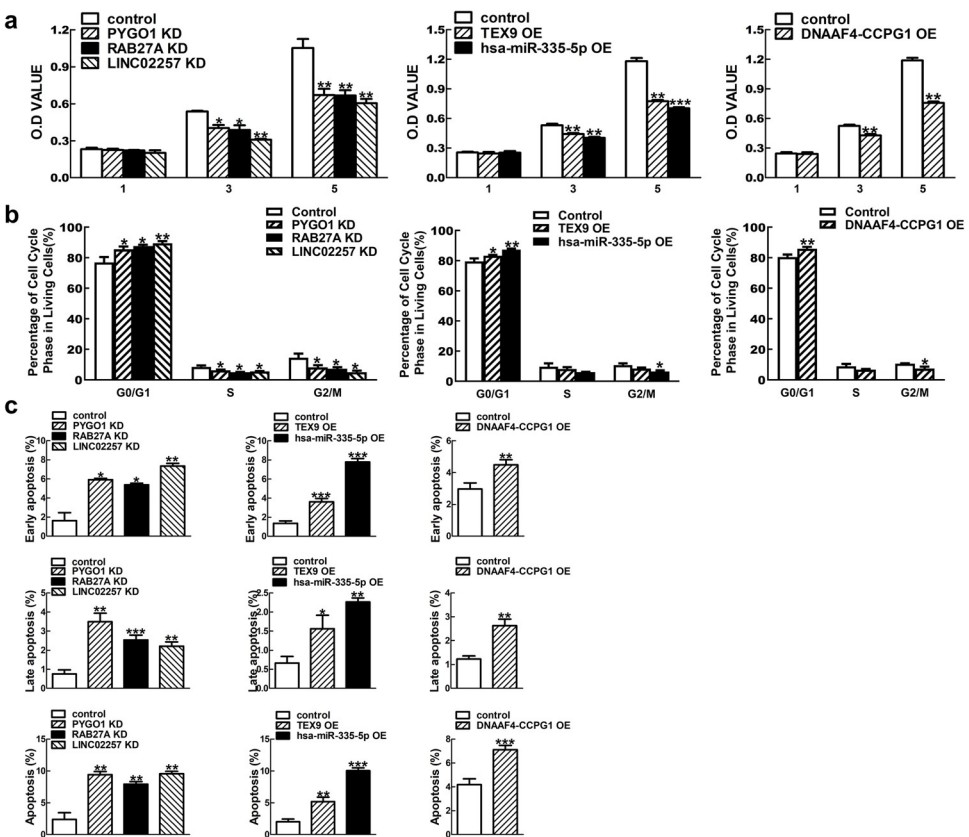

**Fig 6. Loss-of-function and gain-of-function assays inhibited cell proliferation, induced cell cycle arrest at G0/G1 phase and promoted cell apoptosis.** Knockdown of PYGO1, RAB27A, LINC02257 and overexpression of TEX9, hsa-miR-335-5p, DNAAF4-CCPG1 inhibited KFs proliferation after culture for 3 and 5 days (A), arrested cell cycle at G0/G1 phase after culture for 4 days (B) and promoted early, late and total apoptosis of KFs after culture for 4 days (C). Each assay was repeated in at least 3 independent cell samples. (* P < 0.05, ** P < 0.01, *** P < 0.001).

normal skin [25]. The mechanism regarding keloid lesion origin and development is extremely complicated and genetic factors are increasingly recognized as an important contributor to the disease.

Keloid disorder is considered to be a genetically heterogeneous disease that most often shows autosomal dominant inheritance [26]. Familial heritability and prevalence were also found in keloid patients, which supported the concept of genetic susceptibility in keloid disorder [27]. Furthermore, gene mutations such as P53 and FAS were detected in keloid tissues and KFs by whole-exome sequencing [28,29], leading to the impression that genetic mutations in the coding region may cause abnormal function of corresponding proteins and dysregulation of related signalling pathways.

Therefore, large-scaled GWAS analyses of keloid patients were performed to detect the susceptibility loci and genes. Unexpectedly, no gene mutation was found in the exon regions, although a large sample size was involved [2,3]. In addition, a study of 4 keloid disorder cases with family heredity history also failed to reveal obvious exon mutation upon GWAS examination [30]. These negative results imply that gene mutation in exon region might not be a major genetic contributor. Unfortunately, GWAS analysis might be an efficient way to investigate the genetic mechanism of various types of diseases only when enrolling several hundreds to thousands or more participants [31–33].

In recent years, epigenetic inheritance and gene mutation in the non-coding regulatory regions were postulated as important mechanisms for tumour pathogenesis. Somatic mutations instead of germline mutations were found to play a key pathogenic role in squamous cell cancer, prostate cancer and pediatric embryonal tumours [34–36]. In line with the literature reports of somatic mutations in keloid tissues such as P53 and FAS [28,29], no systemic inherited mutation has been found thus far, suggesting that epigenetic abnormality caused by regulating locus within the intron region might be a true genetic contributor to keloid lesion origin and development.

Dislike GWAS, RNA-seq, a widely used assay for epigenetic abnormality analysis, usually requires a relatively small sample size to precisely dissect the genetic and epigenetic mechanism of a certain disease. For example, RNA-seq analysis of the primary mammary epithelial cells (n = 3) and MEKO tumour cell lines (n = 3) found that activation of JNK pathway could prevent tumour initiation [37]. RNA-seq analysis of six mice (3 normal mice and 3 diet-induced obesity mice) elucidated Clec11 regulated proliferation and lipid metabolism in islets [38]. RNA-seq revealed that the genes expression involved in glycolysis was reduced in K562 cells treated with 2 μM Imatinib compared to those of vehicle control (n = 2) which was important in the control of leukemia cell metabolism [39].

Due to the obvious advantages, we conducted an exploratory try of RNA-seq using a self-control model that could minimize individual variation and was able to detect uniformly differentially expressed RNAs. This model has been previously applied for analysing diseases like breast cancer, Paget's disease and malignant melanoma [4–7]. In this study, three cases of severe keloid lesions and their adjacent normal skins were collected to form 3 paired tissue samples for RNA extraction and high-throughput sequencing, which revealed tremendous amounts of DERNAs between paired keloid tissues and nearby normal skins (S1 Fig), and KEGG analysis also showed significantly enriched signalling pathways (Fig 2B).

Some were previously reported in the literature including Wnt signalling pathway and PPARγ signalling pathway [40]. Previous functional analysis of KF gene expression in 32 samples also suggested ECM-receptor interaction pathway was significantly enhanced in keloid lesions [41], which is in agreement with our findings. This study also revealed some signalling pathways that were not reported in the literature for keloid disorder pathogenesis including tyrosine metabolism, pancreatic secretion, neuroactive ligand-receptor interaction,

metabolism of xenobiotics by cytochrome P450, inflammatory mediator regulation of TRP channels, Hippo signalling pathway, estrogen signalling pathway and CAMs, etc. (Fig 2B). These findings might be able to reveal the novel mechanism of keloid disorder. For example, CAMs regulated the fibrotic process in scleroderma [42]. Hippo pathway could modulate pericardial inflammation and myocardial fibrosis [43]. Estrogen signalling pathway and metabolism of xenobiotics by cytochrome P450 were involved in regenerating liver functions [44], and cytochrome P450 abnormality was observed in our keloid tissue single-cell sequencing analysis. The exact roles played by these factors in keloid disorder are worth investigating in the future.

Epigenetics involves the regulation of highly networked genes in which miRNA and lncRNA make a significant contribution. Thus, bioinformatic analysis was performed on DERNAs to form miRNA-mRNA and lncRNA-mRNA network and a total RNA regulation network was built as shown in S4 Fig. Interestingly, the miRNA-mRNA network and the total RNA regulation network showed that hsa-miR-335-5p was a key player as it dominantly regulated a variety of molecules potentially for keloid disorder pathogenesis (Figs 3A, S3 and S4 and S7 Tables), indicating that hsa-miR-335-5p oriented regulation network might be an important part of keloid disorder mechanism, which deserves further exploration.

To better select potential targeting molecules, three modules derived from WGCNA analysis were further jointly analysed with the total RNA regulation network (S5 Fig), which revealed a more complex network (Fig 3B). Among them, a large number of pathways were significantly enriched via this bioinformatic analysis, including PPAR, MAPK, axon guidance, ERBB, insulin, ECM-receptor interaction, Wnt and CAMs pathways (Fig 3C). More importantly, quite some pathways identified via this approach were overlapped with those enriched via simple RNA sequence analysis as shown in Fig 2B including PPAR, ECM-receptor interaction, Wnt and CAMs pathways, suggesting that total RNA regulation network analysis exhibited more advantages over simple mRNA analysis. Among which, MAPK, PPAR and Wnt pathways have already been reported to be closely related with the onset of keloid disorder [21,40]. However, axon guidance, insulin, ERBB, CAMs and ECM-receptor interaction pathways are the first reports by this study and verified by WB (Fig 3D).

Although mutations in the coding region were not captured by GWAS, the function of locus mutation in the intron region remains an open question in the mechanistic investigation of keloid disorder pathogenesis [2,3]. Hi-C was usually used to identify the targeting genes and genetic variants that were influenced by nearby genes via investigating the chromosome interactions, which is beneficial for understanding the mechanism of GWAS variants and their influence on disease pathology [45]. Hi-C can also detect the chromosome interactions and identify the TADs that harbour the risk SNPs. TADs are self-interacting genomic regions, within which DNA sequences prefer to interact with each other more frequently than with sequences outside. They are sub-architectural units of the folded chromatin in the overall 3D genome architecture, which could set a background for gene regulation and be involved in disease susceptibility. Previously, researchers discovered the important pathways associated with the mechanism of disease susceptibility by mapping risk SNPs to TADs regions [46]. Thus, it is important to identify TADs as well as their target genes and risk SNPs located at the particular TADs [45].

We therefore mapped the Hi-C data of Fibroblast IMR90 to the reported non-coding loci (1q41, 3q22.3 and 15p21.3) found in Japanese and Chinese populations suffering from keloid disorder [2,3]. Interestingly, DEmRNA and DElncRNA in TADs located at 1q41 and 15p21.3 (sites shared by Japanese and Chinese) were found, whereas no obvious findings in locus 3q22.3 (unique to Japanese) as shown in Fig 4 [3]. More importantly, the identified ten DERNAs were verified in five other paired keloid lesion-normal skin tissue samples (Fig 5).

Meanwhile, the cytological study showed that overexpressing the RNAs that were downregulated in keloids, and knocking-down the RNAs that were upregulated in keloids could inhibit KF overproliferation through altering the cell cycle distribution and accelerating the cell apoptosis (Figs 6 and S7 and S8). We thus preliminarily verified the feasibility and efficacy of combining GWAS, Hi-C and RNA-seq techniques to elucidate genetic contribution to the pathogenic mechanism of keloid disorder.

In this study, hsa-miR-335-5p was found significantly decreased in keloid lesions. Previously, reduced expression of hsa-miR-335-5p was shown to promote cancer progression, while increasing the expression could effectively inhibit disease progression [47]. As shown, the functional analyses also demonstrated that hsa-miR-335-5p overexpression could inhibit KF proliferation, alter cell cycle distribution and accelerate cell apoptosis (Figs 6 and S7 and S8). However, it remains unclear why the Hi-C TADs analysis failed to reveal differentially expressed hsa-miR-335-5p, suggesting that keloid tissue-specific Hi-C database must be established in order to truly dissect the complexity of keloid disorder genetic mechanism.

Despite the time consumption and intensive efforts made in the past 30 years to try to identify the genetic contribution to keloid disorder pathogenesis, up to now no valuable exon mutation has ever been detected by GWAS, which is in contrast to general expectation because the tumour-like appearance of keloid lesions implies that mutation in the coding region of important molecules should exist, and this expectation has been validated in numerous tumour types [48–50]. This unexpected phenomenon indicates that the traditional approach of genetic investigation using GWAS analysis for exon mutation might be inappropriate and probably has been misled by the tumour-like appearance of keloid lesion, or keloid disorder genetic mechanism might be different from that of cancers although both may share some similarities.

Genetic mutation plays key roles in cancer development, and animal models have been successfully established for many cancer types *in vivo* using cancer cell injection. Different from cancer, keloid lesion animal model never truly succeed and KFs usually lost their pathological phenotype after three passages of *in vitro* culture. These phenomena indicate that epigenetic abnormality, rather than genetic mutation, might be the major contributing way to keloid lesion development. Latest advances in tumour studies using combined approaches of genetic and epigenetic investigation as well as functional evaluation of intronic mutation also provided insightful directions.

Therefore, in this study, RNA sequence and Hi-C technology were combined to dissect the potential mechanism. Although large-scaled GWAS analysis on 714 keloid patients and 2944 controls revealed gene mutation at the loci 1q41 and 15p21.3 without functional evaluation [3], this study did discover ten DERNAs, which were verified in total five paired tissue samples and their functions on cell proliferation, cell cycle and apoptosis in other keloid samples as shown in Figs 5 and 6. It is expected that more pathological factors would be explored when KF specific Hi-C data could be well established.

For the first time, this exploratory study showed that RNA-seq combined with Hi-C and published gene mutation data at the non-coding region could truly reveal many known and unknown discoveries that were further verified by cell functional assays, qPCR and IHC and biochemistry analysis. It also successfully demonstrated the functionality of genetic mutation in the non-coding regions of previous GWAS reports by mapping Hi-C data to the special loci. Such large amounts of pathological findings revealed by this assay indicate that combined Hi-C, RNA-seq and GWAS techniques may represent a correct and efficient approach for investigating the genetic mechanism of keloid disorder pathogenesis. With the increased sample size, establishment of keloid lesion-specific Hi-C database and rapid advancement of sequencing technology, the genetic mechanisms of keloid disorder, a disease with tumour-like pathological changes, will be inevitably dissected.

In summary, this study presents a novel approach of combined analyses of GWAS, RNA-seq and Hi-C to dissect the genetic mechanism of keloid disorder, which is highly efficient and is able to enrich important pathways and identify differentiated molecules that are involved in keloid lesion pathogenic mechanism. This novel approach may also likely be used to explore genetic mechanism of other diseases that are not possible with GWAS analysis alone.

## Materials and methods

### Ethics statement

This study was approved by the Ethics Committee of Shanghai Ninth People's Hospital (2016.8.1–01) with written consent from all keloid tissue donors.

### Keloid patients and specimens

Three keloid patients (named K1, K2 and K3) diagnosed with severe keloid lesions (as shown in Fig 2A) were recruited. Then three pathological specimens from the outer zone of keloid lesions as well as three adjacent normal skins were collected during keloid lesion resection surgery. The distance between the keloid lesions and normal skins was about 1 cm. In addition, other five paired tissue samples (named KT1-KT5) were used for the verification of DERNAs and corresponding proteins using IHC, WB and RT-qPCR. Besides, nine keloid tissue samples (named KS1-KS9) were also employed for in vitro cell function analysis of the DERNAs. All the tissue samples were donated by keloid patients for research purposes only with written informed consent and the information of keloid tissue samples is listed in S1 Table.

### Analysis of DERNAs

mRNA-seq, lncRNA-seq and miRNA-seq analyses were respectively performed on the six samples (three keloid lesions and three corresponding adjacent normal skins) by Genminix Informatics co., Ltd (Shanghai, China) as previously described [51,52]. Package DESeq2 was used to compare the DERNAs between paired keloid lesion and normal skin samples of all the three keloid patients. The DEmRNAs and DElncRNAs with the adjusted P value < 0.05 and |log2FC| ≥ 2 based on DESeq2 analysis were selected. Regarding miRNAs, the extracted counts were normalized across all samples using DESeq2 package, and miRNAs with the adjusted P value < 0.05 and |log2FC| ≥ 0.67 were selected as DEmiRNAs. Hierarchical cluster heatmap representing the expression direction and intensity of DEGs, DElncRNAs and DEmiRNAs was generated using the pheatmap (R package) (https://cran.r-project.org/web/packages/pheatmap/index.html) based on Euclidean distance. Functional enrichment analysis based on the KEGG functional hierarchy was performed on DEGs, target genes regulated by DElncRNAs and target genes regulated by DEmiRNAs (see Fig 1).

### Identification of co-expression modules by performing clustering analysis

Pearson's correlation coefficient (PCC) was calculated between expressed values of each lncRNA-mRNA pair in keloid lesion and normal skin samples. The co-expressed lncRNA-mRNA pairs with PCC > 0.9 were selected as significant interaction pairs [53]. Furthermore, co-expression analysis was performed using WGCNA to identify the modules of high correlation and the P value of DEGs and DElncRNAs between keloid lesion and normal skin samples was calculated and regarded as a feature of each gene module [54]. The clustering analysis was conducted to identify significant co-expression modules. Then the significant modules were selected and related lncRNA-mRNA-miRNA interaction network was constructed (see Fig 1).

## Construction of miRNA-target network

MiRTarBase is a comprehensive database of experimentally verified miRNA-target interactions, which were verified either by highly reliable methods such as WB, fluorescence reporter experiments, RT-qPCR or predicted by high-throughput experiments like photoactivatable ribonucleoside enhanced crosslinking, immunoprecipitation and crosslinking immunoprecipitation coupled with high throughput sequencing [55]. Therefore, it was used to explore the functions of DEmiRNAs. In order to investigate the roles of DEmiRNAs in regulating the expression of target genes, miRNA-target interaction network was constructed based on the DEmiRNA-target interaction pairs using miRanda and TargetScan database, and the nodes in the DEmiRNA-target network were ranked according to a descending order of each degree by performing topological characteristic analysis (see Fig 1) [56].

## Construction of lncRNA-associated RNA network

The RNA network consisting of DEmiRNA-DEG-DElncRNA interactions was constructed based on the relationship information of DEmiRNA-DEG interaction pairs and DElncRNA--DEG interaction pairs. The network was displayed using Cytoscape software [57].

## Functional enrichment analysis

Functional enrichment analyses were conducted on the DEGs, the targeting genes regulated by DElncRNAs, and the targeting genes regulated by DEmiRNAs in the lncRNA-mRNA-miRNA network constructed by the significant gene modules. The network diagram was built and the DEmRNAs/DEmiRNAs/DElncRNAs enriched in KEGG biological pathway were analysed by network analyst tools (P < 0.01 by Fisher's exact probability test) [58].

## Combined analyses of Hi-C and RNA-seq data

Considering that fibroblast proliferation is the main pathogenic mechanism of keloid disorder, the genome-wide chromosome conformation of Human Fibroblast IMR90 captured by Hi-C (http://promoter.bx.psu.edu/hi-c/view.php) was obtained and analysed. The selected reference genome was Hg38, and resolution ratio was set as 40k [59]. Afterwards, the Hi-C RNA expression of the TADs in these three susceptible loci (1q41, 3q22.3 and 15p21.3) found in Japanese and Chinese populations with keloid disorder were mapped to the keloid tissue RNA-seq data. And DERNAs in these three TADs were selected for further validation through WB, IHC and RT-qPCR assay. All the antibody information is listed in S2 Table.

## WB assay

Important molecules were selected based on the KEGG enrichment data, Hi-C and RNA-seq joint analysis data. These selected molecules were subjected to WB to examine the expression levels of encoded proteins of five genes including HHIPL2, NEDD4, PYGO1, RAB27A and TEX9. WB was also used to examine the representative proteins involved in the eight signalling pathways including LAMA1, LAMA2 and LAMA3 in ECM–receptor interaction, IGF1R in insulin signalling pathway, PPARγ in PPAR pathway, AREG, ERBB1, ERBB3 in ERBB pathway, Slit2 in axon guidance pathway, and L-selectin, ICAM1 in CAMs signalling pathway. The analyses were performed as follows. Briefly, about 20 μg whole protein lysate was loaded onto a 4–20% SDS-PAGE gel (Bio-Rad, Hercules, CA). The fractionated proteins were transferred onto a nitrocellulose membrane (Bio-Rad, Hercules, CA), which was subsequently blocked with 5% bovine serum albumin for 1 hour. The membranes were incubated at 4˚C overnight with the primary antibodies. Anti-β-actin was used as a loading control. The membranes were

thoroughly washed in washing buffer and incubated with the secondary antibodies for 1 hour at room temperature. The protein bands were eventually visualized using an enhanced chemiluminescence detection kit (Thermo Scientific, 32106). All the antibody information is listed in S2 Table.

## IHC assay

Keloid tissue and adjacent normal skin tissue sections were incubated with anti-HHIPL2, anti-NEDD4, anti-PYGO1, anti-RAB27A and anti-TEX9 antibodies at 4°C overnight followed by two washes in phosphate-buffered saline (PBS). Then, the sections were incubated with a secondary antibody at 37°C for 30 min and washed in PBS, stained with 3,3'-diaminobenzidine chromogenic kit. Afterwards, the samples were counterstained with hematoxylin and rinsed with distilled water. After being dried, they were observed under a microscope and digitally recorded. Integral optical density (IOD)/area values were used to assess the expression level in the immunohistochemically stained tissue sections with the image analysis program Image-Pro Plus 6.0 in three randomly selected fields.

## RNA extraction and RT-qPCR

The expression levels of mRNAs and lncRNAs were validated by RT-qPCR. Total RNA was extracted from the keloid tissue and adjacent normal skin tissues of other five randomly selected keloid patients using EZ-press RNA Purification Kit (EZBioscience, USA). A 4×Reverse Transcription Master Mix (EZBioscience) was used for reverse transcription reaction at 42°C for 15 min, 95°C for 3 min. The 2×SYBR Green qPCR Master Mix (EZBioscience) was used to perform qPCR as follows: initial denaturation at 95°C for 5 min, followed by 40 amplification cycles (10s at 95°C for denaturation, and 30s at 60°C for annealing and extension) using a Strata Gene Mx3000p (Applied Biosystems). Each interested gene was normalized to the housekeeping gene *GAPDH* and the fold change was compared relatively to that of the control sample. qPCR assay was performed in triplicate and repeated at least three times.

## Isolation and culture of human KFs

KFs were obtained from keloid patients underwent keloid lesion excision procedure. The experimental protocol was approved by the Ethics Committee of Shanghai Ninth People's Hospital. As previously reported [60], the keloid tissues of inflammatory outer zone were excised and the epidermis was removed. The remaining dermis was cut into small fragments and digested with 0.3% collagenase (SERA, Heidelberg, Germany) in Dulbecco's modified Eagle's medium (DMEM) (Hyclone, Logan City, UT) for 6 h at 37°C on a rotator. Afterwards, the cell suspension was centrifuged and resuspended in DMEM with 10% fetal bovine serum (FBS, Gibco, Grand Island, NY). The cells were seeded onto a 10-cm culture dish (BD, Falcon, TX) at a regular density ($1.0×10^6$ per dish) and passaged by 0.25% trypsin-EDTA (Ethylene Diamine Tetraacetic Acid) (Gibco) when they reached 80% confluency.

## The overexpression and knockdown of genes and non-coding RNAs

PYGO1, RAB27A, LINC02257 shRNA plasmids, and hsa-miR-335-5p, TEX9 and DNAAF4-CCPG1-sgRNA overexpression plasmids were respectively co-transfected with the lentiviral packaging plasmids (Genomeditech, China) into 293 T cells [61]. The virus-containing supernatants were collected at 48 h after transfection and filtered using a 0.45 μm cellulose acetate filter (Merck Millipore, USA). Then the supernatants were diluted 2 times with serum-free DMEM containing polybrene (YEASEN, China) at a final concentration of 5 μg/ml. The

mixed solutions were added to the KFs (seeded onto 6-well culture plates at $1\times10^5$ per well) for another 16 hours of incubation before the exchange with fresh DMEM culture medium. After another 72 hours of incubation, the stably transfected cells were selected with 1 μg/ml puromycin (Sigma, USA). KFs transfected by lentivirus carrying negative control under the same conditions were used as controls.

### Cell viability assay

The control and transfected KFs were cultured in 96-well plate at 2000 cells/well for 5 days. Cell Counting Kit 8 (CCK-8; Beyotime Institute of Biotechnology, Haimen, China) was used to examine cell viability at 1, 3- and 5-day post-transfection. Results are presented as the cell proliferation (%) relative to control cells.

### Cell cycle analysis

The control and transfected KFs were seeded in 6-well culture plates at $1\times10^5$ cells per well for 4 days, then the cells were collected and fixed in 70% ice-cold ethanol at 4°C overnight. Afterwards, they were washed with PBS, incubated in PBS containing 100 μg/ml propidium iodide (PI), RNase (1 mg/ml), and 0.1% Triton X-100 at 4°C for 30 min. The proportion of KFs in cell cycle distributions was measured by the Epics Altra Flow Cytometer equipped with Modi-Fit LT v2.0 software (Verity Software House, Topsham, ME, USA) in half an hour.

### Cell apoptosis analysis

The control and transfected KFs cultured for 4 days were harvested with 0.25% trypsin followed by two washes in PBS, and centrifuged at 1500 rpm for 5 min. Then the cells were washed with ice-cold PBS, and resuspended in 195 μl of binding buffer (cat. no. 556454, BD Biosciences, USA). After being stained with 5 μl Annexin V-FITC and 10 μl PI in dark for 20 min at 37°C, cell samples were analysed with flow cytometry.

### Statistical analysis

For the part of bioinformatics analysis, Package DESeq2 was used to compare the DERNAs between paired keloid lesions and normal skins of all three keloid patients. The DEmRNAs and DElncRNAs with a P value < 0.05 and $|log2FC| \geq 2$ were selected. The DEmiRNAs with a P value < 0.05 and $|log2FC| \geq 0.67$ were selected. The co-expressed lncRNA-mRNA pairs with PCC > 0.9 were selected as significant interaction pairs [53]. The differential RNAs of module oriented total network in KEGG pathway were analysed by network analysis tools (P < 0.01 by Fisher's exact probability test). The DERNAs within susceptibility loci were selected based on the criteria of P < 0.05 and $|log2FC| \geq 1$.

For the part of *in vitro* experiments, the statistical analyses were processed using the statistical software Statistical Package for the Social Sciences (version 19.0, Chicago). The data are represented as the mean ± standard derivation. Student's t-test was employed to analyze the differences between control and experimental counterparts, and P < 0.05 was considered statistically significant.

## Supporting information

**S1 Fig. Clustering of the RNA-seq data from three paired tissue samples.** Heatmap showed DEGs, DElncRNAs and DEmiRNAs between keloids and their adjacent normal skins.
(TIF)

**S2 Fig. DElncRNA-target gene interaction pairs were modelled into 7 modules using WGCNA.**
(TIF)

**S3 Fig. Construction of DEmiRNA-target gene interaction pairs.**
(TIF)

**S4 Fig. Construction of total DERNAs regulation network.**
(TIF)

**S5 Fig. Mapping the three modules into the total DERNAs regulation network.**
(TIF)

**S6 Fig. Mapping the eight signalling pathways of the three modules into the total DERNAs regulation network.**
(TIF)

**S7 Fig. Cell cycle was arrested at G0/G1 phase after gene knockdown and overexpression.** Representative histograms of cell cycle phase distribution of KFs with knockdown of PYGO1, RAB27A, LINC02257 (A) and overexpression of TEX9, hsa-miR-335-5p (B) and DNAAF4-CCPG1 (C).
(TIF)

**S8 Fig. Loss-of-function and gain-of-function assays promoted cell apoptosis.** Representative apoptosis pattern of KFs with knockdown of PYGO1, RAB27A, LINC02257 (A) and overexpression of TEX9, hsa-miR-335-5p (B) and DNAAF4-CCPG1 (C).
(TIF)

**S1 Table. Demographic data of keloid samples used in this study.**
(XLSX)

**S2 Table. Antibodies used in western blotting and immunohistochemical staining.**
(XLSX)

**S3 Table. The differentially expressed mRNAs between keloids and corresponding adjacent normal skins.**
(XLSX)

**S4 Table. The differentially expressed lncRNAs between keloids and corresponding adjacent normal skins.**
(XLSX)

**S5 Table. The differentially expressed miRNAs between keloids and corresponding adjacent normal skins.**
(XLSX)

**S6 Table. The whole DEmiRNAs targeted mRNAs.**
(XLSX)

**S7 Table. The topological features of top 20 nodes in the miRNA-target network.**
(XLSX)

**S8 Table. The risk SNPs in the three patients.**
(XLSX)

**S1 Data. Numerical data underlying graphs.**
(XLSX)

## Author Contributions

**Conceptualization:** Wei Liu.

**Data curation:** Jia Huang.

**Formal analysis:** Jia Huang.

**Funding acquisition:** Wei Liu.

**Investigation:** Xiaobo Zhou.

**Methodology:** Jia Huang.

**Project administration:** Wei Liu.

**Resources:** Wenbo Wang, Zhen Gao, Xiaoli Wu.

**Software:** Xiaobo Zhou.

**Supervision:** Guangdong Zhou, WenJie Zhang, Wei Liu.

**Visualization:** Jia Huang.

**Writing – original draft:** Jia Huang.

**Writing – review & editing:** Wei Liu.

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
