## [Decision Letter · Decision Letter 0]

6 Feb 2022

Dear Dr Prof. Liu:

Thank you very much for submitting your Research Article entitled 'Combined analyses of RNA-sequence and Hi-C along with GWAS loci — A novel approach to dissect keloid genetic mechanism' to PLOS Genetics.

The manuscript was fully evaluated at the editorial level and by independent peer reviewers. The reviewers appreciated the attention to an important topic but identified some concerns that we ask you address in a revised manuscript

We therefore ask you to modify the manuscript according to the review recommendations. Your revisions should address the specific points made by each reviewer, and provide a detailed list of your responses to the review comments and a description of the changes you have made in the manuscript.

Yours sincerely,

Michael Tirgan, MD

Guest Editor

PLOS Genetics

Chengqi YI

Section Editor: Methods

PLOS Genetics

Dear Prof. Liu:

We thank you for submitting your manuscript to PLOS Genetics. Your Manuscript was review by reviewers. Please respond to the reviewer comments and revise and resubmit your manuscript.

Thank you

PLOS Genetics

Reviewer's Responses to Questions

**Comments to the Authors:**

Reviewer #1: This manuscript describes an RNASeq approach combined with Hi-C analysis comparing 3 keloids with adjacent normal skin pairs and investigates the GWAS loci previously described by Nakashima et al., 2010. The authors identified several potential pathways leading to keloids and identified several genes within these pathways that showed differential expression by immunohistochemistry and performed over-expression and knock-down experiments to show effects on cell survival.

Overall, I recommend that authors revisit English grammar and sentence structure in the manuscript for better readability.

L48: Please rephrase sentence as keloids don’t just appear “only” due to re-occurrence.

L49 and L270: Do you mean the control of keloid growth and recurrence is not well understood? Please rephrase.

L50: replace “exon gene mutation” with “exonic gene mutations”

L53: replace with “by mutations in regulatory”.

L56: replace with “may lead to a breakthrough in the discovery of”. Do you mean further analyses are needed? Please rephrase.

L68: recommended change to plural “gene mutations are increasingly recognized as one of the critical risk factors”.

L82: change to “For example, overexpression….” as this lncRNA is being used as an example and has nothing to do with keloids.

General comment: Were the differentially expressed genes detected in all samples uniformly or was there heterogeneity (all keloids versus all controls in your various analyses)? This needs to be addressed upfront and also appropriately addressed in the Discussion section. For better reading, the relevant paragraphs could mention the keloid samples that were used for specific experiments [e.g., K1-K3 (Suppl. Table S1)].

L133: hsa-miR-355-5p was down-regulated in your RNA-Seq experiments. Were the interacting genes that are listed here also DEGs in your screen? This is not clear from your sentence.

L256: Were these biological and technical replicates? Was the knock-down performed with keloid samples used for RNASeq or in some of the additional keloid samples (KS1-KS9)? Was there any heterogeneity seen in the latter samples or did all samples have a comparable knock-down result? Please expand for more clarity.

L366: The fact that the samples K1-K9 are keloid-normal sample pairs needs to be described in the Methods section. Fig. 5 does not show or explain that these samples were used. It would be clearer if transcript abundance relative to a housekeeping gene would be presented in box plots (Livak and Schmittgen, Methods, 2001; PMID: 11846609). Thus, it would also be more clear whether a message is highly expressed or at low levels.

L453: WGCNA appears to be best suited to study linear relationships in a larger number of samples. You analyzed each of 3 keloids to the corresponding normal skin sample - or all samples together? Please describe your methodology and the limitations of WGCNA.

L516: “cultivated” do you mean incubated?

L513: Since TEX9 showed the opposite effect from Western blot results the methods section should describe how you did control stainings for IHC. Isotype controls?

Fig.1: This is an interesting presentation of a summary but may be easier to read as a flow chart.

Fig. 2: Please spell out WGCNA in the figure legend. What do colors in 2c stand for? Needs to be mentioned somewhere.

Fig 3D: The Western bands are presumably not from a single blot, yet only one housekeeping band is shown. Can you quantify the keloid/normal scar bands to their respective housekeeping gene?

Fig 5D: It would be clearer if transcript abundance relative to a housekeeping gene would be presented in box plots. Thus, it would also be more clear whether a message is highly expressed or at low levels.

Reviewer #2: This study focuses on keloid disorder, a group of fibroproliferative tumors, keloids being the prototype. These studies are welcome in the context of the notion that there are no effective or specific treatments for this disorder. The study is a methodological tour-de-force combining several state of the art technologies, including GWAS, RNA-seq and Hi-C, analyses, and the data are subjected to extensive bioinformatics, which has generated a large number of data which have been meticulously catalogue in this manuscript.

There were a number of issues at that should be considered by the authors for improvement of the study.

1. The study focuses on keloids but only three individual keloid samples, together with the adjacent normal skin, were analyzed. The author's description of the keloids is somewhat incompletes in that the authors do not make a clear distinction between keloids and hypertrophic scars. In fact, the description in the first paragraph of the Introduction is more applicable to hypertrophic scars.

Furthermore, there is no histological documentation of the type of keloid tissue that is being analyzed. The importance of this point is emphasized by the fact that keloid lesions can be highly heterogeneous, some parts of the lesions being highly cellular with a large number of inflammatory cells, while other parts are characterized by dense fiber meshwork of extracellular matrix of connective tissue, mostly collagen, with a few myofibroblastic cells embedded into it. Thus, it is not clear how representative their findings are in the spectrum of keloid pathology.

2. While the genetic contribution to keloid development is well established through population studies, the authors state that “genetic contribution is well expected due to the presence of autosomal dominant inheritance.” In fact, autosomal dominant inheritance has bee documented only in a very few families. In the present manuscript, there is no reference to family history of keloids in the three patients studied.

3. Additional studies were performed using cultured keloid fibroblasts Isolated from apparently inflammatory outer zone of the lesions. The pitfall of these experiments relate to the culture conditions of the fibroblasts. First, the cells were cultured on plastic culture dishes, but it has been well documented that the culture of keloid fibroblasts under such standard fibroblast culture conditions results in loss of the collagen overproducing phenotype, and the keloid fibroblasts in culture do not recapitulate the characteristics of cells in keloid lesions in situ. Consequently, it is very unclear if the changes observed in the molecular studies are really representative of the characteristics of these cells when part of the keloid lesions.

4. Another deficiency of the culture conditions is the apparent lack of ascorbic acid (vitamin C) which is an obligatory requirement for the synthesis of not only collagens but also a number of other proteins, and the corresponding mRNA transcript levels have been shown to be altered by addition of vitamin C to the tissue culture medium.

5. The bioinformatics of the data derived from the molecular analyse, I.e., GWAS, RNA-seq and Hi-C, has resulted in a large number of data that are faithfully detailed in the manuscript. It is apparent, however, that many of the pathways identified to be altered in keloid samples vs. control skin may have very little, if anything, to do with the pathomechanisms of keloids. Thus, while technically for the most part correct, these studies have provided information which is very difficult to correlate to the development and growth of keloids, with little therapeutic implications for this, currently intractable disorder. As indicated by the authors, this is really an exploratory study with a limited number of samples and an increase in sample size and advancements in sequencing technologies are required to dissect the pathological changes associated with keloids.

Reviewer#3

The authors have undertaken a very important step in better understanding the genetics of keloid disorder.  This work deserves to be published; however, this manuscript can be much improved prior to being published.  Please consider the following:

1- The title of the manuscript - *Keloid-specific gene expression profiling* – will easily attract the readers who are familiar with keloid disorder and are interested in this topic, therefore there is no need to provide an exhaustive review of what a scar is, or what a keloid is. Furthermore, there is no general agreement on the definitions that are provided for scar, hypertrophic scar, etc.

I would encourage the authors to remove much of introduction and start their abstract from where they state: “In an effort to find out keloid-specific genes---” and the introduction from where they state: “To date, there have been five reports---".  Those interested in this publication are very familiar with the underlying disorder.

2- Authors refer to Keloids as Scars. Keloids are not scars. The authors have somewhat alluded to this in the context of their manuscript. The authors should more precisely define different pathologies, and perhaps consider the following:

Consider using the term “Keloid Disorder” to refer to the underlying genetic disorder that keloid patients have.

Consider using the term “Keloid Lesion” to refer to the actual lesions.

Consider using the term “Keloid Tissue” to refer to the tissue of a keloid lesion.

Consider using the term “Keloid Patient” to refer to the patient who suffers from keloid disorder.

Avoid using the term “Keloid” alone, as using this term alone leaved the reader to decide is the author is talking about the Keloid Disorder or the Keloid Lesions.

Avoid using “exaggerated scars such as hypertrophic scars and keloids” Just state hypertrophic scars and keloid lesions.

3- The core of the introduction starts where the authors state: “To date, there have been five reports comparing gene expression patterns in keloid and related scar lesions—”

Authors should consider using a phrase such as:  To our knowledge, there have been five reports comparing gene expression patterns in keloid tissue.

Also, “related scar lesions” is vague. What do you mean by related scar lesions? In the same patient, in other patients.

4 - In the introduction section, the authors state: “In this study, we aimed to compare the overall gene expression patterns of reversible and irreversible scars and provide insight into the molecular and cellular mechanisms behind these lesions.”

Please revise the phrase “irreversible scars such as keloids and reversible scars.” This is way too confusing.

Tell us what you exactly did.

5- In the introduction section, the authors state: “abnormal fibroproliferative scar lesions,” 

The authors use several different terms, and it is unclear what they mean by these. For instance, in here, do you mean Keloid Lesions?

6- In the result section, the authors stated that” To reveal the gene expression characteristics of keloids and related lesions, we collected 4 keloid tissues, 5 hypertrophic scars, and 3 normotrophic scars.”

Were these tissue samples taken from ONE patient? Or from 12 different patients?  Authors need to better describe the origin of the tissues that they analyzed. Did 4 keloids come from 4 different patients or from one keloid patient? Same as the other tissues.

**Have all data underlying the figures and results presented in the manuscript been provided?**

Reviewer #1: **No: **I did not find link to data sets generated for this study. No statement of data sharing has been made. The Nakashima et al., 2010 data set is publicly available.

Reviewer #2: Yes

PLOS authors have the option to publish the peer review history of their article (what does this mean?). If published, this will include your full peer review and any attached files.

Reviewer #1: No

Reviewer #2: No

---

## [Editor Report · Decision Letter 1]

25 Mar 2022

Dear Prof. Liu:

We are pleased to inform you that your manuscript entitled "Combined analyses of RNA-sequence and Hi-C along with GWAS loci — A novel approach to dissect keloid disorder genetic mechanism" has been editorially accepted for publication in PLOS Genetics. Congratulations!

Yours sincerely,

Michael Tirgan, MD

Guest Editor

PLOS Genetics

Chengqi YI

Section Editor: Methods

PLOS Genetics

Comments from the reviewers (if applicable):

**Data Deposition**

http://datadryad.org/submit?journalID=pgenetics&manu=PGENETICS-D-21-01585R1

**Press Queries**

---

## [Editor Report · Acceptance letter]

27 Apr 2022

PGENETICS-D-21-01585R1 

Combined analyses of RNA-sequence and Hi-C along with GWAS loci — A novel approach to dissect keloid disorder genetic mechanism 

Dear Dr Liu, 

We are pleased to inform you that your manuscript entitled "Combined analyses of RNA-sequence and Hi-C along with GWAS loci — A novel approach to dissect keloid disorder genetic mechanism" has been formally accepted for publication in PLOS Genetics! Your manuscript is now with our production department and you will be notified of the publication date in due course.

With kind regards,

Livia Horvath

PLOS Genetics

On behalf of:
